# Predictive Factors of Poor Outcome in Sanders Type III and IV Calcaneal Fractures Treated with an Open Reduction and Internal Fixation with Plate: A Medium-Term Follow-Up

**DOI:** 10.3390/jcm11195660

**Published:** 2022-09-26

**Authors:** Luigi Cianni, Raffaele Vitiello, Tommaso Greco, Mattia Sirgiovanni, Giulia Ragonesi, Giulio Maccauro, Carlo Perisano

**Affiliations:** 1Department of Ageing, Neurosciences, Head-Neck and Orthopedics Sciences, Orthopedics and Trauma Surgery Unit, Fondazione Policlinico Universitario Agostino Gemelli IRCCS, 00168 Rome, Italy; 2Orthopedics and Trauma Surgery, Università Cattolica del Sacro Cuore, 00168 Rome, Italy; 3Department of Pneumology, Allergology and Intensive Care Medicine, University of Saarland, 66421 Homburg, Germany

**Keywords:** calcaneus fractures, fracture, ORIF, plate fixation, Sanders III–IV classification

## Abstract

Background: Consensus on the treatment for severely comminuted calcaneus fractures has yet to be found. This study aims to analyze the functional and radiological short- and medium-term outcomes of displaced calcaneus fractures of type III and IV treated with ORIF, and to identify, if present, the early predictors of unfavorable outcomes. Methods: Thirty-three calcaneal fractures were included, 23 type III and 10 type IV, according to Sanders classification. AOFAS scales for ankle and hindfoot and SF-12 were used. Böhler and Gissane angles were analyzed before and after surgery. Results: The minimum follow-up was six years. The mean AOFAS score at six months was 16.5 points (24.2 ± 10.8 vs 10.8 ± 9.5; *p* = 0.03) with better outcomes in patients with Sanders type III fractures. This difference decreased in the subsequent follow-up. Likewise, the mental and physical score of SF-12 had the same trend. Two wound infections and no deep infections were recorded in the Sanders type III fracture group. Instead, in the Sanders type IV group, there were four wound infections and one deep infection. Conclusions: Clinical and radiological outcomes in Sanders Type III and Type IV calcaneus fractures treated with plate and screws were very similar in long-term follow-up. If ORIF provided better short- to medium-term follow-up in Sanders type III fracture, these benefits have been lost in six years. Polytrauma and psychiatric patients showed significantly lower clinical outcomes in long-term follow-up, appearing as the most reliable negative predictors.

## 1. Introduction

Calcaneus fractures account for 1–4% of all fractures (60% of all tarsal fractures), most being intra-articular and mostly due to trauma from falling from a height; there is a prevalence in male patients with a ratio of 2.4 points in the third decade of life, with women affected mainly in the postmenopausal age [1]. Although there is a lack of agreement in the literature on the treatment of choice for displaced intra-articular calcaneal fractures, open reduction with internal fixation (ORIF) and percutaneous surgery (PS) are the most common surgical procedures [2,3]. According to the Sanders classification system, which evaluates calcaneus fractures based on the number of fracture lines and their position on coronal CT images, identifying four types of increasing severity, poor functional outcomes are frequently described for type III and IV, with consequent significant impairment of the patient’s quality of life, regardless of the chosen therapeutic approach [4]. Many authors argue the need for a surgical approach rather than a non-operative one, as it would ensure better long-term outcomes and a lower risk of post-traumatic subtalar arthritis and secondary subtalar arthrodesis [5,6,7,8,9]. Some argue that primary subtalar arthrodesis combined with open reduction and internal fixation (ORIF) is a successful method of achieving good anatomical and functional results and should be fully considered [10,11]. This study aims to analyze the short- and medium-term outcomes of displaced calcaneus fractures of type III and IV treated with ORIF with plate and screws, assessing functional outcomes, the overall impact on the quality of life and the anatomical reduction based on the restoration of the Gissane and Böhler angles. The goal is to identify, if present, the early predictive factors of poor outcome.

## 2. Materials and Methods

All patients with calcaneal fractures type III or IV according to Sanders classification, treated with ORIF with plate, at our institute between January 2010 and December 2014 who met the inclusion criteria were included in the study. Patients were divided into two groups according to the Sanders classification: type III and type IV. All patients gave informed consent to the study prior to the start of the data analysis. All procedures performed were in accordance with the 1964 Helsinki Declaration and its subsequent amendments. The study design was approved by the Orthopedic Department council and our school board. As this is an approval from the Review Board of Orthopedic and Traumatology Institute, there is no code. The approval date is the session of 24 May 2022. Informed consent was obtained from all individual participants included in the study. The inclusion criteria were patients with Sander type III or IV calcaneal fracture, ORIF treatment with plate and screws. The exclusion criteria were other treatments than ORIF with plate and screws and follow-up of less than six years.

### 2.1. Surgical Technique

All fractures included in the study were surgically treated by two experienced trauma surgeons. In all cases, general anesthesia or peripheral anesthesia was performed. All patients received Cephazoline 2 g i.v. as an antibiotic prophylaxis before surgery, if not contraindicated. A lateral approach was used. A tourniquet was placed at the root of the limb. Seligson’s surgical approach was always used: after debridement and open reduction and temporary fixation, an anatomical calcaneal plate (LCP–Synthes © plate) was applied [12]. The use of the augmentation was performed according to the surgeon’s preference. A splint was placed after surgery. The sutures were removed after two to three weeks. The splint was removed on day 40 of surgery and active range of motion of the ankle was initiated. The transition from no weight-bearing to partial weight-bearing was allowed at eight weeks; walking without aids and total weight-bearing was achieved at 12 weeks. Patients were followed up regularly at two and four weeks after surgery and then every three months for the first two years, then annually. Starting four weeks after surgery, an x-ray was taken at each clinical evaluation to assess bone healing; when complete bone healing was achieved, an annual radiograph was taken.

### 2.2. Clinical Evaluation

Demographic data were collected on patients such as sex and age, number and type of comorbidities (when present), smoking habits, and return to physical activity; in addition, the time between injury and surgery, the no weight-bearing period and postoperative complications were reported. The AOFAS ankle and hindfoot scale was used to assess clinical outcomes [13]; this scale spans from 0 (no abnormality) to a maximum of 100 points and evaluates pain, function and alignment. The 12-Item Short Form Health Survey (SF-12) was used to measure the impact of a calcaneus fracture on quality of life. SF-12 is a generic scale that evaluates the physical and mental status of patients [14]. All complications (wound dehiscence, deep infection or neurological damage) were recorded during hospitalization and outpatient follow-up. Wound dehiscence or surgical site infection was defined as surgical wound healing with the presence of redness, edema and discharge in the absence of deep tissue involvement or general symptoms [15].

### 2.3. Radiological Assessment

Fractures were diagnosed through a standard radiographic series in each case (anteroposterior, lateral and axial views). Computed tomography (CT) was performed to classify the fractures according to the Sanders classification. Radiographic images were analyzed before and after surgery by measuring Böhler and Gissane angles on the lateral projections. The Böhler angle normally varies between 25° and 40°. The crucial angle of Gissane typically measures between 120° and 145° [16].

### 2.4. Data Analysis

GraphPad QuickCalcs (GraphPad Software, San Diego, CA, USA) was used for data analysis. Data were reported as mean and standard deviation (+SD). An unpaired T test was used to compare anthropometric, anamnestic, radiological and clinical data (AOFAS and SF-12). Non-continuous variables were analyzed through the chi-square test. Significance was set for *p* < 0.05.

## 3. Results

During the study period, 47 patients with 55 calcaneal fractures (36 type III and 19 type IV according to Sanders) were admitted in our department. Of these patients, two died before the calcaneal fractures were treated as they were polytrauma patients with major associated injuries. Of the remaining 45 (a total of 52 fractures, 35 type III and 17 type IV), 11 patients (nine type III e 4 type IV) were excluded due to the type of treatment received: seven patients were treated conservatively and four were managed with an external fixator.

Of the remaining 34 patients with a total of 39 fractures (26 type III and 13 type IV), six patients (3 type III and 3 type IV) were excluded, as they did not complete the six-year follow-up.

Twenty-eight patients with 33 calcaneus fractures were enrolled according to the inclusion and exclusion criteria. These included 23 type III and 10 type IV according to the Sanders classification. There were 21 males and 12 females, and the mean age was 52.5 years (±18.6). The minimum follow-up was six years, and the mean was 7.24 years (±1.6). Demographic data, smoking habits, length of stay and days prior to surgery were reported in Table 1. No statistical differences were found in demographic data between Sanders type III or IV, except for the number of polytraumas, mental illnesses and length of stay, which were more frequent in Sanders’ type IV group.

In Sanders type IV group, 40% of patients also reported other fractures: 12% ribs, 12% upper limb (humerus, radius, clavicle), 10% lower limb (patella, Lisfranc and femur), 3% facial bone and 3% vertebrae.

In four patients in Sanders type IV group and in only one patient in Sanders type III group, augmentation was used (Triosite^®^ Bioactive Ceramic Bone Graft Substitute), and no clinical or radiological differences were found among these two groups.

The mean Gissane angle before surgery was 128.6° (±14.3°) and after surgery it was 122.1° (±16.1°), the mean Böhler angle before the surgery was 19.6° (±7.9°) and after surgery was 24.1° (±10.3°), therefore with a complete restoration of the normal angle. Gissane and Bohler angles were restored after surgery, but no statistical differences were found between them in Sanders type III and IV before and after the surgery (Table 2).

The mean AOFAS score at six months was 16.5 (±11.8) points (24.2 ± 10.8 vs 10.8 ± 9.5; *p* = 0.03) with better outcomes in patients affected by Sanders type III fractures. This difference gradually decreased in the next follow-up (AOFAS one year 38.3 ± 17.2 vs 32.6 ± 13.4 *p* = 0.4; AOFAS six years 49.2 ± 17.5 vs 56 ± 10.8 *p* = 0.3) with the tendency to inversion between the two groups but without statistical significance (Figure 1). Likewise, SF-12 Mental and Physical score had the same trend (Figure 2 and Figure 3).

In the Sanders type III group, two wound infections occurred (2/23 registered patients; 8%) and no deep infections were recorded. Four wound infections (4/10 patients; 40%) and a deep infection (1/10 patients; 10%) occurred in the Sanders type IV group. The superficial infections, after consultation with the infectious pathologist, were treated with advanced dressings and targeted antibiotic therapy. For the deep infection, as the fracture had already healed, hardware removal, bone curettage and targeted antibiotic therapy were performed.

In the Sanders type III group, three patients (3/23 patients; 13%) required hardware removal surgery for discomfort, while in the Sanders type IV group seven patients required the surgery (7/10 patients; 30%). All 10 patients had the hardware removed at least one year after the initial surgery. Statistical difference between the two groups was reached (*p* = 0.001).

## 4. Discussion

The main finding of the present study is that the clinical and radiological outcomes in Sanders type III and type IV calcaneus fractures treated with ORIF were very similar at medium-term follow-up. If the ORIF provided better short- to medium-term follow-up in Sanders type III fractures, these advantages were lost over a six-year follow-up.

The literature shows conflicting results on the outcomes of calcaneus fractures and the potential superiority of one treatment option over another or conservative treatment in those types of fractures [7,17,18,19,20].

Our study reveals, in agreement with other authors, that the higher grade of comminution found in type III and IV fractures has a strong prognostic impact. This can be partially explained by the energy of the trauma (leading not only to a complex displaced fracture, but also to a substantial loss of integrity of the joint cartilage) and by the technical difficulties in achieving anatomical reduction [21,22,23]. In addition, if the angles of Gissane and Böhler were restored after surgery, the clinical results were poor. Makki et al. found an association between restoration of the Bohler angle and a better outcome, since in the group with a restored Bohler angle there were no poor results, while 33% of fractures with a Böhler angle < 30° were associated with poor outcomes [24].

This appears to be inconsistent with our work, as the AOFAS score was not significantly different between patients with and without a restored Böhler angle, which did not seem to correlate with a better outcome.

Sanders et al. described the clinical and radiographic outcomes of 132 displaced intra-articular type II, III and IV calcaneus fractures treated by the lateral approach, lag screw fixation of the joint, body H-plate fixation and without augment. While the clinical outcome in 73% of type II fractures was classified as good or excellent (13% ultimately required subtalar fusion due to damaged cartilage despite anatomic reduction), 20% and 73% of types III and IV led to complete clinical failures, respectively (3). Our study showed a more severe medium-term outcome for type III fractures, with 80% poor outcomes and 20% fair outcomes, but not for type IV fractures, with 60% poor outcomes and 40% fair outcomes. Good results were not observed in either follow-up.

Although our short-term follow-up showed significantly different pain and functional levels between patients with type III and IV fractures, mid-term follow-up scores tend to level off. Neither the short-term nor the medium-term average score can be considered a favorable outcome. This also applies to the SF-12 scale, the results of which in both follow-ups are an expression of a below-average state of health.

When ORIF was used as a treatment option for intra-articular calcaneus fractures, several complications were described, along with a significant number of re-operations. Wound infections were the most frequent, mainly wound dehiscence (12.1%), superficial infection (3.2% to 18.1%) and deep infection (up to 13.6%). Our results are in line with literature data (18% superficial infection and 3% deep infection). Risk factors for wound complications include smoking, diabetes, open fractures, and high body mass index [13,25]. De Groot et al. underline that these complications didn’t influence the clinical outcome in the medium and long term, and our study had results that are in line with the literature [26].

A higher risk of wound dehiscence and/or infection was found in polytrauma patients (50% risk vs 20% in case of simple trauma). Statistically significant differences were described by comparing the pre-surgical Böhler angles in polytrauma and non-polytrauma patients (13.6° ± 6.4° vs 23° ± 6.6°; *p* = 0.004), confirming a more severe presentation between polytrauma. However, these differences tended to decrease and lose statistical significance in the postoperative period, which is the basis of a good surgical procedure. Polytrauma patients often present with Sanders type IV fractures and other fractures due to the etiology of trauma due to mental illness and suicide attempts. This condition explains the length of hospital stay, which was significantly longer among patients with type IV fractures than among those with type III fractures (9.8 and 31 days, respectively).

In the work of Renovell-Ferrer et al., the mean AOFAS score was lower in the polytrauma group than in the non-polytrauma group, but not significantly lower, as there was no statistical association between polytrauma patients and second surgeries, including subtalar arthrodesis [27]. The same authors found a significant statistical relationship between polytrauma patients and those with psychiatric comorbidities, severe trauma or severe injuries [27]. Patients with psychiatric comorbidities had significantly worse health-related quality of life but no clinical outcome compared to people without this background. In our study, a correlation can be found between polytrauma and psychiatric patients, as 57.14% of the polytrauma patients had some psychiatric comorbidities and 100% of those with a psychiatric background came to the emergency room following suicide attempt. Furthermore, the presence of psychiatric comorbidities correlated with significantly lower physical scores of AOFAS (45 vs. 54; *p* = 0.04) and SF-12 (31.1 vs. 38.92; *p* = 0.05).

The two most important predictors of poor clinical outcomes in our sample were polytrauma and mental illness. In these patients, different surgical treatments, such as external fixation or conservative treatment, could be considered, yielding similar clinical results [18,23].

In light of the data obtained, patients with Sanders type III and IV calcaneus fractures generally have poor medium-term outcomes; therefore, less invasive surgical methods, such as external fixation [28,29,30] or minimally invasive tibiotalocalcaneal arthrodesis [31], with a low rate of complications, could be favored in this type of patient. Comparative studies will be needed to better investigate this aspect.

Our study has several limitations: first of all, the sample size was small (particularly in the Sanders IV group) and the population was heterogeneous. Although it was not the primary aim of the study, another limitation was the lack of a control group for comparison with conservatively treated patients. We presented the AOFAS score (an unvalidated functional scale) in only three follow-ups over six years (six months, one year, six years) and only recorded early complications. Our minimum follow-up was six years, but a longer period was needed.

## 5. Conclusions

Sanders type III and IV calcaneal fractures still pose a challenge to the orthopedic surgeon, because they are often associated with poor clinical outcomes and poor health-related quality of life. The optimal management of these fractures is even more difficult, considering the lack of consensus as to which approach is best to use across different clinical backgrounds.

ORIF treatment in Sanders type III calcaneus fractures provides better short-term outcomes than Sanders type IV, but these benefits were lost in mid-term follow-up.

Despite what was initially expected, the anatomical reconstruction defined through the restoration of the Böhler and Gissane angles does not appear to significantly influence the clinical outcomes. Polytrauma and psychiatric patients showed significantly lower clinical outcomes in long-term follow-up, appearing as the most reliable negative predictors.

## Figures and Tables

**Figure 1 jcm-11-05660-f001:**
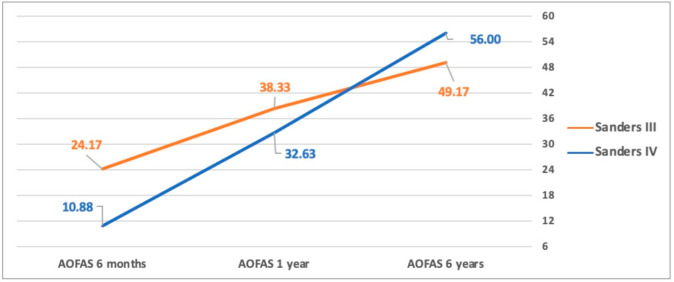
AOFAS score between Sanders types III and IV. The AOFAS score at six months follow-up showed better outcomes in patients affected by Sanders type III fractures (*p* = 0.03). This difference gradually decreases at one-year follow-up with the tendency to inversion between the two groups, but without statistical significance.

**Figure 2 jcm-11-05660-f002:**
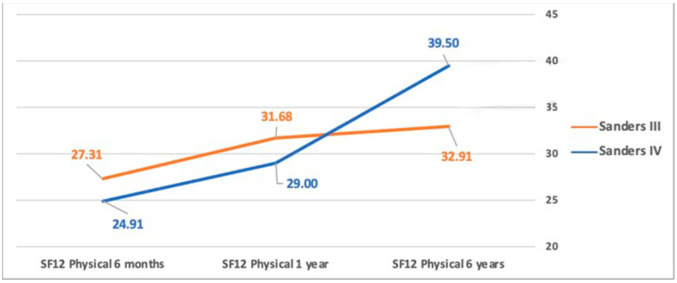
Physical SF-12 score between Sanders types III and IV. Similarly, the AOFAS score at six months follow-up showed better outcomes in patients affected by Sanders type III fractures. This difference gradually decreases at one-year follow-up with the tendency to inversion between the two groups, but without statistical significance.

**Figure 3 jcm-11-05660-f003:**
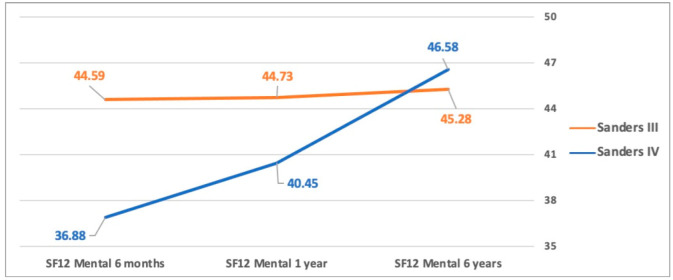
Mental SF-12 score between Sanders types III and IV. Similarly, the AOFAS score at six months follow-up showed better outcomes in patients affected by Sanders type III fractures. This difference gradually decreases at one-year follow-up with the tendency to inversion between the two groups, but without statistical significance.

**Table 1 jcm-11-05660-t001:** Baseline characteristics.

	Sanders III	Sanders IV	Global	*p*
Sex	15 M–8 F	6 M–4 F	21 M–12 F	
Age (year) ^1^	55.2 ± 15	49 ± 19.6	52.5 ± 18.6	0.4
Follow-up (year) ^1^	7.0 ± 1.2	7.4 ± 1.9	7.2 ± 1.6	0.7
Length of stay (days) ^1^	9.8 ± 12.8	31 ± 24.5	22.5 ± 22.7	0.01
Bilaterality ^2^	1	4	5	
Days before surgery ^1^	15.9 ± 12.6	16.5 ± 7.3	16.2 ± 9.5	0.8
Polytrauma ^2^	0	8	8	0.001
Mental Illness ^2^	0	6	6	0.02
Smoking ^2^	3	3	6	
Hardware removal ^2^	3	7	10	0.001

In brackets measurement unit. M: male; F: female. ^1^ Data were reported as mean ± SD. ^2^ Data were reported as absolute value.

**Table 2 jcm-11-05660-t002:** Radiographic characteristics.

	Sanders III	Sanders IV	Global	*p*
Gissane angle before surgery (degrees) ^1^	122.6 ± 13.7	133.6 ± 13.3	128.6 ± 14.3	0.07
Gissane angle after surgery (degrees) ^1^	118.4 ± 16.2	125 ± 16.1	122.1 ± 16.1	0.3
Gissane angle variation (degrees) ^1^	−4.1 ± 16.2	−10.3 ± 41.5	−0.8 ± 33.3	0.6
Böhler angle before surgery (degrees) ^1^	22.4 ± 7.6	17.2 ± 7.6	19.6 ± 7.9	0.1
Böhler angle after surgery (degrees) ^1^	25.1 ± 10.5	23.4 ± 27.1	24.1 ± 10.3	0.7
Böhler angle variation (degrees) ^1^	2.7 ± 5.9	7.4 ± 11.1	4.9 ± 9.1	0.2

In brackets measurement unit. ^1^ Data were reported as mean ± SD.

## Data Availability

The study data will be available upon request to the corresponding author.

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
