# Peer review of "Predictive Factors of Poor Outcome in Sanders Type III and IV Calcaneal Fractures Treated with an Open Reduction and Internal Fixation with Plate: A Medium-Term Follow-Up"

_jcm, 2022, doi:10.3390/jcm11195660_

Round 1

Reviewer 1 Report

Many thanks to the authors for having presented a so interesting retrospective study about “Predictive factors of poor outcome in Sanders type III and IV calcaneal fractures treated with an open reduction and internal fixation with plate: a medium-term follow-up “.

Please before resubmitting the revision version of the article, read the editorial rules carefully, and check for other editorial aspects, such as: text alignment, text justification at the head, etc. The language is so good that the manuscript does not need to be corrected by a person of English mother tongue.

 Abstract

The abstract is well structured, it contains the main information and result of the study. 

Key words

Please provide them in alphabetic order.

 Introduction

The introduction is well structured. It well explains the aim of the study. However, I do not agree with the authors about this sentence (lines 35-7): A consensus has yet to be found on treatment options for severely comminuted calcaneus fractures, for which the literature does not have conclusive studies and does not support the absolute superiority of one method over another [2]. Please, modify it quoting:

·         J Orthop Surg Res. 2016 Aug 22;11(1):92. doi: 10.1186/s13018-016-0426-6. Radiographic and functional outcomes after displaced intra-articular calcaneal fractures: a comparative cohort study among the traditional open technique (ORIF) and percutaneous surgical procedures (PS)

 Methods

This section contains enough information to understand and possibly repeat the study. It well defines inclusion and exclusion criteria, the sources of the patient, the period of recruitment and follow-up period. However, this section does not reflect the Strobe Statement-Checklist for cohort studies. Please, remove any results from this section and move them into Results one. In particular, the number of patients with Sanders type IV (10) is not appropriate to justify the results: quote it in the study limit section.

No ethical statements from ethical committee or review board are described. Please, provide the name of the local ethical committee and the number and date of study approval in this section also.  

 Results

The results presented are quite complete, but the number of patients does not reflect the MM section (in MM section you talk about 33 patients, in this section you talk about 28 patients with 33 calcaneus fractures). Please modify this section according Strobe guidelines and clarify the number of patients and fractures in the cohort.

The results are reproducible and reflective of clinical expectations. They are displayed in a readable fashion. In this section are also well defined the complications that occurred in the two groups.

 Discussion

The length and content of the discussion communicate the main information of the paper. It recognizes the limitations of the study.

Line 199: “Wound infections are the most frequent, mainly wound dehiscence (12.1%), superficial infection (3.2% to 18.1%) and deep infection (up to 13.6%). Our results are in line with literature data (18% superficial infection and 1% deep infection)” ….However, your deep infection results are not in line with literature data or those percentual values are not right. Please clarify this section and correct them!

Further, the results are not adequately discussed about the most recent percutaneous and minimally invasive techniques for subtalar or double arthrodesis of the hindfoot. Please, discuss these aspects and quote:

·         Minimally Invasive Surgery for Tibiotalocalcaneal Arthrodesis Using a Retrograde Intramedullary Nail: Preliminary Results of an Innovative Modified Technique. J Foot Ankle Surg. 2016 Nov-Dec;55(6):1130-1138. doi: 10.1053/j.jfas.2016.06.002. Epub 2016 Aug 11.

Limitations section are described. However, AOFAS scale was not a validated, please report this further limit of the study.

Conclusions

The conclusions reflect and refer to the results and the methods of the study.

References

The references are quite up to date. Please delete those before 2000 replacing them with newer ones and adding those suggested previously.

Figures

The number and quality of figures are appropriate to transmit the main information of the paper.

Author Response

Many thanks to the authors for having presented a so interesting retrospective study about “Predictive factors of poor outcome in Sanders type III and IV calcaneal fractures treated with an open reduction and internal fixation with plate: a medium-term follow-up “.

Please before resubmitting the revision version of the article, read the editorial rules carefully, and check for other editorial aspects, such as: text alignment, text justification at the head, etc. The language is so good that the manuscript does not need to be corrected by a person of English mother tongue.

Thanks to the reviewer for the careful and detailed review.

Below is a point-by-point answer.

 Abstract

The abstract is well structured, it contains the main information and result of the study. 

Key words

Please provide them in alphabetic order.

Answer: Thanks for the suggestion. The keywords have been changed as suggested.

 Introduction

The introduction is well structured. It well explains the aim of the study. However, I do not agree with the authors about this sentence (lines 35-7): A consensus has yet to be found on treatment options for severely comminuted calcaneus fractures, for which the literature does not have conclusive studies and does not support the absolute superiority of one method over another [2]. Please, modify it quoting:

  • J Orthop Surg Res. 2016 Aug 22;11(1):92. doi: 10.1186/s13018-016-0426-6. Radiographic and functional outcomes after displaced intra-articular calcaneal fractures: a comparative cohort study among the traditional open technique (ORIF) and percutaneous surgical procedures (PS)

Answer: Thanks for the suggestion. The sentence has been modified and the reference added.

Methods

This section contains enough information to understand and possibly repeat the study. It well defines inclusion and exclusion criteria, the sources of the patient, the period of recruitment and follow-up period. However, this section does not reflect the Strobe Statement-Checklist for cohort studies. Please, remove any results from this section and move them into Results one. In particular, the number of patients with Sanders type IV (10) is not appropriate to justify the results: quote it in the study limit section.

Answer: Thanks for the suggestion. The section has been edited as suggested, the changes are highlighted in yellow in the text. The limitations have also been changed as suggested (line 247).

No ethical statements from ethical committee or review board are described. Please, provide the name of the local ethical committee and the number and date of study approval in this section also.  

Answer: Thank you for the remark.

We have further specified the ethical evaluation of our manuscript by stating the following sentence

"The study design was approved by the board of the Department of Orthopaedics and our school board. As this is an approval by the Review Committee of the Orthopaedic and Traumatology Institute, no code is provided. The date of approval is the meeting of 24 May 2022”.

 Results

The results presented are quite complete, but the number of patients does not reflect the MM section (in MM section you talk about 33 patients, in this section you talk about 28 patients with 33 calcaneus fractures). Please modify this section according Strobe guidelines and clarify the number of patients and fractures in the cohort. 

The results are reproducible and reflective of clinical expectations. They are displayed in a readable fashion. In this section are also well defined the complications that occurred in the two groups.

Answer: Thank you for your comment and notes. There was indeed a mistake in the typing, for which we apologise. In the MM section, the number of patients has been removed (in accordance with Strobe guidelines). As reported on line 111, the total number of patients was 28, with 33 calcaneal fractures in total.

 Discussion

The length and content of the discussion communicate the main information of the paper. It recognizes the limitations of the study.

Line 199: “Wound infections are the most frequent, mainly wound dehiscence (12.1%), superficial infection (3.2% to 18.1%) and deep infection (up to 13.6%). Our results are in line with literature data (18% superficial infection and 1% deep infection)” ….However, your deep infection results are not in line with literature data or those percentual values are not right. Please clarify this section and correct them!

Further, the results are not adequately discussed about the most recent percutaneous and minimally invasive techniques for subtalar or double arthrodesis of the hindfoot. Please, discuss these aspects and quote:

Minimally Invasive Surgery for Tibiotalocalcaneal Arthrodesis Using a Retrograde Intramedullary Nail: Preliminary Results of an Innovative Modified Technique. J Foot Ankle Surg. 2016 Nov-Dec;55(6):1130-1138. doi: 10.1053/j.jfas.2016.06.002. Epub 2016 Aug 11.

Answer: Thank you for the suggestion that allows us to expand on the discussion. There was a typo in the percentage of deep infections, which has been corrected from 1% to 3%. This result is reasonably in line with the literature, also considering the small numbers in our study. Thank you for the proposed reference, which was duly added and discussed.

Limitations section are described. However, AOFAS scale was not a validated, please report this further limit of the study.

Answer: we have updated the limitations of the study.

Conclusions

The conclusions reflect and refer to the results and the methods of the study.

Answer: Thank you for the comment.

References 

The references are quite up to date. Please delete those before 2000 replacing them with newer ones and adding those suggested previously.

Answer: Thank you for your suggestion, which will certainly enhance the quality and validity of the manuscript.

We have removed or replaced all pre-2000 references, leaving only one that represents a historical article on the management of calcaneal fractures. 

Figures

The number and quality of figures are appropriate to transmit the main information of the paper.

Answer: Thank you for the comment.

Reviewer 2 Report

Dear Authors, 

I have a few issues to be taken care of in this interesting paper.

1. Please correct  the number of patients, you stated that there were 28 patients and then stated that there were 21 males and 12 females. Please attend to it.

2. You showed that 10 patients needed the hardware removal. What were the reasons and when did it happen in the follow up?

3. Did you have any reoperations in this cohort of patients?

4. How did you treat the mentioned superficial and one deep infection?

5. It would be interesting to compare operated Sanders 3 and 4 patients with the patients from your Institute that were treated conservatively.

Best regards 

Author Response

Dear Authors, 

I have a few issues to be taken care of in this interesting paper.

  1. Please correct the number of patients, you stated that there were 28 patients and then stated that there were 21 males and 12 females. Please attend to it.

Answer: Thank you for your comment and notes. There was indeed a mistake in the typing, for which we apologise. In the MM section, the number of patients has been removed (in accordance with Strobe guidelines). As reported on line 111, the total number of patients was 28, with 33 calcaneal fractures in total.

  1. You showed that 10 patients needed the hardware removal. What were the reasons and when did it happen in the follow up?

Answer: Thank you for your question, which allows us to better specify the results of our study (as added in the text in lines 144-45). The 10 patients who removed the implants all did so because of intolerance to the implants and discomfort, and all after at least one year after treatment.

  1. Did you have any reoperations in this cohort of patients?

Answer: Thanks for the question. Re-operations were performed in nine cases for implant removal and in one case (of deep infection) for hardware removal and bone curettage.

  1. How did you treat the mentioned superficial and one deep infection?

Answer: Thank you for the very pertinent question. We have further elaborated and specified our management of superficial and deep infections in lines 138-42.

  1. It would be interesting to compare operated Sanders 3 and 4 patients with the patients from your Institute that were treated conservatively.

Answer: The comment is very interesting and had already been considered by us. At the moment, we are not able to have such a large case series of conservatively treated patients to make a comparison, due to our predominant indication for ORIF.

Best regards

We thank the reviewer for a very careful and relevant review and hope, with our answers, that we have answered all his questions and satisfied him.

Round 2

Reviewer 1 Report

The authors answered my comments properly, improving manuscript quality.
well done!

Author Response

Thank you for your comment.